# Laminar flow substantially affects the morphology and functional phenotype of glomerular endothelial cells

**Daan C. 't Hart, Johan van der Vlag**[ID]ᵒ**, Tom Nijenhuis**\*ᵒ

Department of Nephrology, Radboud Institute for Molecular Life Sciences (RIMLS), Radboud University Medical Center, Nijmegen, The Netherlands

ᵒ These authors contributed equally to this work.
\* Tom.Nijenhuis@radboudumc.nl

**Citation:** 't Hart DC, van der Vlag J, Nijenhuis T (2021) Laminar flow substantially affects the morphology and functional phenotype of glomerular endothelial cells. PLoS ONE 16(5): e0251129. https://doi.org/10.1371/journal. pone.0251129

**Data Availability Statement:** The data files supporting the findings in this study can be found at the Open Science Framework database at the following DOI: 10.17605/OSF.IO/XY9S6.

## Abstract

Shear stress induced by laminar blood flow has a profound effect on the morphology and functional phenotype of macrovascular endothelial cells. The influence of laminar flow on the glomerular microvascular endothelium, however, remains largely elusive. The glomerular endothelium, including its glycocalyx, is a crucial part of the glomerular filtration barrier, which is involved in blood filtration. We therefore investigated the influence of laminar flow-induced shear stress on the glomerular endothelium. Conditionally immortalized mouse glomerular endothelial cells were cultured for 7 days under a laminar flow of 5 dyn/cm² to mimic the glomerular blood flow. The cells were subsequently analysed for changes in morphology, expression of shear stress-responsive genes, nitric oxide production, glycocalyx composition, expression of anti-oxidant genes and the inflammatory response. Culture under laminar flow resulted in cytoskeletal rearrangement and cell alignment compared to static conditions. Moreover, production of nitric oxide was increased and the expression of the main functional component of the glycocalyx, Heparan Sulfate, was enhanced in response to shear stress. Furthermore, glomerular endothelial cells demonstrated a quiescent phenotype under flow, characterized by a decreased expression of the pro-inflammatory gene ICAM-1 and increased expression of the anti-oxidant enzymes HO-1 and NQO1. Upon exposure to the inflammatory stimulus TNFα, however, glomerular endothelial cells cultured under laminar flow showed an enhanced inflammatory response. In conclusion, laminar flow extensively affects the morphology and functional phenotype of glomerular endothelial cells in culture. Furthermore, glomerular endothelial cells respond differently to shear stress compared to macrovascular endothelium. To improve the translation of future *in vitro* studies with glomerular endothelial cells to the *in vivo* situation, it appears therefore crucial to culture glomerular endothelial cells under physiological flow conditions.

**Funding:** This work was financially supported by Kolff Senior Postdoc Career Stimulation Grant 13OKS023 (to T.N.) from the Dutch Kidney Foundation. The funders had no role in study design, data collection and analysis, decision to publish, or preparation of the manuscript.

**Competing interests:** The authors have declared that no competing interests exist.

## Introduction

The nephron is the functional unit of the kidney, in which the glomerulus is responsible for filtration of the blood, a process in which passage of plasma proteins into the urine is restricted in a charge- and size selective manner [1]. The two cell types constituting the glomerular filtration barrier are glomerular endothelial cells (GEnC) and podocytes (glomerular visceral epithelial cells), which are separated by the glomerular basement membrane (GBM) [2]. GEnC are lining the glomerular capillaries, contain numerous fenestrae and are covered by a glycocalyx, which is a thick layer of carbohydrates. The endothelial glycocalyx plays an important role in glomerular function, by contributing to endothelial barrier function, glomerular structural integrity and by preventing immune cell and cytokine adhesion [3–5]. Hyaluronic Acid (HA) is the main non-sulfated glycosaminoglycan (GAG) in the glycocalyx, providing its gel-like structure. Heparan Sulfate (HS) is the main functional sulphated GAG contributing to the charge-selective function of the endothelial barrier. Under pathological conditions, HS within the glycocalyx is degraded by heparanase (HPSE). HS degradation compromises the barrier function of the endothelium, and the glycocalyx transforms into a proinflammatory docking station that facilitates binding of cytokines [3,5–7]. Podocytes are highly specialized epithelial cells that wrap around the glomerular capillary with their foot processes, forming a sieve-like structure between adjacent foot processes, called the slit diaphragm.

Blood flow is well-known to affect the morphology and phenotype of the aortic macrovascular endothelium by inducing shear stress [8–11]. The importance of shear stress on macrovascular endothelium has for example been shown during the pathogenesis of atherosclerosis [12]. When aortic flow is disturbed, this predisposes aortic regions to the development of atherosclerosis and stimulates a pro-inflammatory phenotype of the endothelial cells. In addition, culture of macrovascular endothelial cells under flow for 7 days resulted in the induction of an anti-inflammatory phenotype and reduced oxidative stress [8–11]. Besides inducing this quiescent phenotype, shear stress has been shown to alter the glycocalyx composition and the expression of glycocalyx modifying enzymes of the macrovasculature [13–15].

Based on aforementioned findings in macrovascular endothelium, it is hypothesized that blood flow also influences, among others, the inflammatory phenotype and the endothelial glycocalyx composition of GEnC. By preventing the pro-inflammatory phenotype of GEnC and regulating the glycocalyx composition, shear stress might play a pivotal role in the maintenance of a healthy glomerular endothelium, the preservation of the glomerular filtration barrier and eventually the correct filtration of blood in the kidney. Importantly, GEnC are microvascular endothelial cells with several unique characteristics in comparison with the macrovascular endothelium. For example, GEnC are extremely flattened endothelial cells, contain numerous non-diaphragmed fenestrae, experience lower flow rates *in vivo* and lack the surrounding vascular smooth muscle cells (VSMC) [2]. It is therefore unknown whether the results from studies using macrovascular endothelial cells can be translated to microvascular cells like GEnC. In the very few studies in which GEnC were cultured under laminar flow, the effect of shear stress on the anti-oxidant response, the inflammatory response and the glycocalyx composition was not examined [16–20]. Studies investigating the effect of shear stress on GEnC primarily focussed on the secretion of paracrine factors like endothelin-1 and nitric oxide (NO) [16–19]. In addition, in two studies which investigated the influence of shear stress on GEnC, non-uniform flow was applied to the cells [16,17]. Furthermore, GEnC were solely exposed to shear stress for 1 day in three of the studies [16,18,19]. From studies using macrovascular endothelial cells, it is known that shear stress should be applied for 7 days in a uniform way to accurately mimic the influence of blood flow *in vivo* [8–11]. In addition, non-uniform laminar flow is known to play an important pathogenic role in atherosclerosis [12]. It remains

therefore elusive how laminar flow affects the functional phenotype of GEnC. As cultured GEnCs are often used as experimental tools in research on the glomerular filtration barrier, whether GEnCs are cultured under static or laminar flow conditions could be of importance when translating *in vitro* results to the *in vivo* situation.

To study the effect of shear stress on GEnC in a way that more closely resembles the *in vivo* situation, we compared GEnC cultured for 7 days under laminar flow using physiologically relevant levels of shear stress to the static culture condition [21,22]. In summary, we show that laminar flow for 7 days has a profound effect on the morphology and functional phenotype of GEnC *in vitro*, as evidenced by increased NO production, increased HS expression and an altered anti-oxidant and inflammatory response. In addition, GEnC respond differently to laminar flow than macrovascular endothelial cells, as shown by their changes in glycocalyx composition and response to inflammatory stimulation.

## Materials and methods

### Cell culture

Conditionally immortalized mouse glomerular endothelial cells (mGEnC) were cultured as described previously [23]. This cell line was isolated and characterized by our lab by Rops *et al* in 2004 [23]. Briefly, mGEnC were seeded in μ-slides Luer $I^{0.8}$ (ibidi, Gräfelfing, Germany), which were coated at 37°C for 1hr with 1% gelatine, dried for 1hr at 37°C and subsequently coated for 1hr with 1.5μg/cm$^2$ bovine Fibronectin at 37°C. 3 hours upon seeding, the μ-slides were connected to the ibidi Pump system and mGEnC were exposed to a shear stress of 1 dyn/cm$^2$ for 30 minutes, subsequently to 2 dyn/cm$^2$ for 30 minutes and finally to 5 dyn/cm$^2$ for the remaining 7 days.

### RNA isolation, reversed transcription and quantitative PCR analysis

RNA was isolated from mGEnC using Trizol (Thermofisher Scientific, Breda, The Netherlands) and 0.5–1μg RNA was reverse-transcribed into cDNA using the Transcription First Strand cDNA synthesis kit (Roche, Woerden, The Netherlands) according to manufacturer's instructions. Quantitative gene expression levels were determined by quantitative PCR using SYBR Green (Roche diagnostics, Woerden, The Netherlands) on a CFX 96 C1000 Thermal Cycler (Biorad, Lunteren, The Netherlands) and normalized to Hypoxanthine-guanine phosphoribosyltransferase (HPRT) or glyceraldehyde 3-phosphate dehydrogenase (GAPDH) levels using the delta-delta $C_T$ method. Sequences of gene specific primers are listed in Table 1.

### Immunofluorescent staining

Cells were fixed for 10 minutes with 2% paraformaldehyde (PFA) and 4% sucrose for eNOS and iNOS staining and with 90% ice-cold acetone for wheat germ agglutinin (WGA) and Heparan Sulfate epitope JM403 staining. In case of PFA fixation, cells were permeabilized for 10 minutes with a 0.3% Triton X-100 solution in 1x PBS and subsequently blocked for 30 minutes with blocking solution consisting of 2% BSA, 2% FCS and 0.2% fish gelatine. When cells were fixed with acetone, 1% BSA was used as blocking solution. Primary antibodies or WGA lectin were diluted in blocking buffer and incubated for 1hr at room temperature (RT): eNOS (1:100 dilution, Thermofisher, PA3-031A), Heparan Sulfate (1:100 dilution, Amsbio, JM403), iNOS (1:100 dilution, abcam, ab178945) and biotinylated WGA (1:1000 dilution, vectorlabs). Goat anti-Rabbit Alexa 488, Goat anti-Mouse Alexa 488 and Alexa 488 conjugated Streptavidin (all from Thermofisher Scientific) were diluted in blocking buffer (1:200) and incubated for 45 minutes in the dark at RT. When using Phalloidin-TRITC to stain the actin skeleton,

**Table 1. Overview of the primers used in this study.**

| Gene name | Gene symbol | Forward sequence (5'-3') | Reverse sequence (5'-3') |
|---|---|---|---|
| HPRT | HPRT | TCCTCCTCAGACCGCTTTT | CCTGGTTCATCATCGCTAATC |
| GAPDH | GAPDH | GTGTTCCTACCCCCAATGTGTC | GTGTTCCTACCCCCAATGTGTC |
| KLF2 | KLF2 | ACCAAGAGCTCGCACCTAAA | GTGGCACTGAAAGGGTCTGT |
| CYP1B1 | CYP1B1 | GATGTGCCTGCCACTATTACG | CCCACAACCTGGTCCAACTCA |
| ICAM-1 | ICAM-1 | GTCGAAGGTGGTTCTTCTGAG | TCCGTCTGCAGGTCATCTTAGG |
| HPSE | HPSE | GAGCGGAGCAAACTCCGAGTGTATC | GATCCAGAATTTGACCGTTCAGTTGG |
| iNOS | NOS2 | AATCTTGGAGCGAGTTGTGG | CAGGAAGTAGGTGAGGGCTTG |
| eNOS | NOS3 | GGTAGTTAGGGCATCCTGCTG | GTCTGGGACTCACTGTCAAAG |
| Nrf2 | NRF2 | TAGATGACCATGAGTCGCTTGC | TAGATGACCATGAGTCGCTTGC |
| HO-1 | HO-1 | CCAGTCGCCTCCAGAGTTTC | CAAATCCTGGGGCATGCTGTC |
| NQO1 | NQO1 | CATTGCAGTGGTTTGGGGTG | GCAGGATGCCACTCTGAATC |
| HAS1 | HAS1 | GAGGCCTGGTACAACCAAAAG | CTCAACCAACGAAGGAAGGAG |
| HAS2 | HAS2 | TGGTGAGACAGAAGAGTCCCA | GATGAGGCAGGGTCAAGCAT |

cells were incubated for 45 minutes with Phalloidin-TRITC (1:200, Thermofisher) in 1% BSA and 0.1% sodium azide solution after permeabilization with 0.3% Triton X-100 for 10 minutes.

## Nitric oxide detection

Production of nitric oxide (NO) in mGEnC was visualized using the NO-sensitive dye 4-Amino-5-Methylamino-2',7'-Difluoroscein (DAF-FM) diacetate. Upon 7 days differentiation under either static condition or under a flow of 5 dyn/cm$^2$, cells were incubated for 60 minutes with 10μM DAF-FM diacetate in phenol red free and FBS free medium at 37˚C. Cells were subsequently washed three times with Hank's balanced salt solution (HBSS, Gibco, Breda, The Netherlands) to remove excess probe and incubated for an additional 45 minutes. Immediately thereafter, fluorescent images were made at a Leica DMI600B microscope.

## Statistical analyses

Numerical results are presented as mean ± SEM. ImageJ (v.1.47) was used for the analysis and quantification of fluorescent intensity. Statistical analysis was conducted by a two-tailed student's t-test when 2 experimental groups were compared for qPCR experiments and a one-way ANOVA with a Tukey's Post-Hoc test was used when 3 or more experimental conditions were compared. For the analysis of the fold changes in fluorescent intensity, a paired t-test was conducted. All statistical analyses were performed using GraphPad Prism version 5 (GraphPad Software, Inc., San Diego, CA, USA). A P-value of <0.05 was considered statistically significant.

# Results

## Laminar flow induces morphological changes in GEnC

The most profound alterations of macrovascular endothelial cells in response to shear stress are the alignment of the cells and the rearrangement of the actin cytoskeleton in the direction of the laminar flow [24–26]. We therefore investigated how the morphology and the actin skeleton alignment of GEnC changed in response to laminar flow. Culture under laminar flow for 7 days resulted in the partial alignment of GEnC in the direction of the flow (**Fig 1A**). In addition, GEnC cultured under static conditions displayed a random distribution of the actin

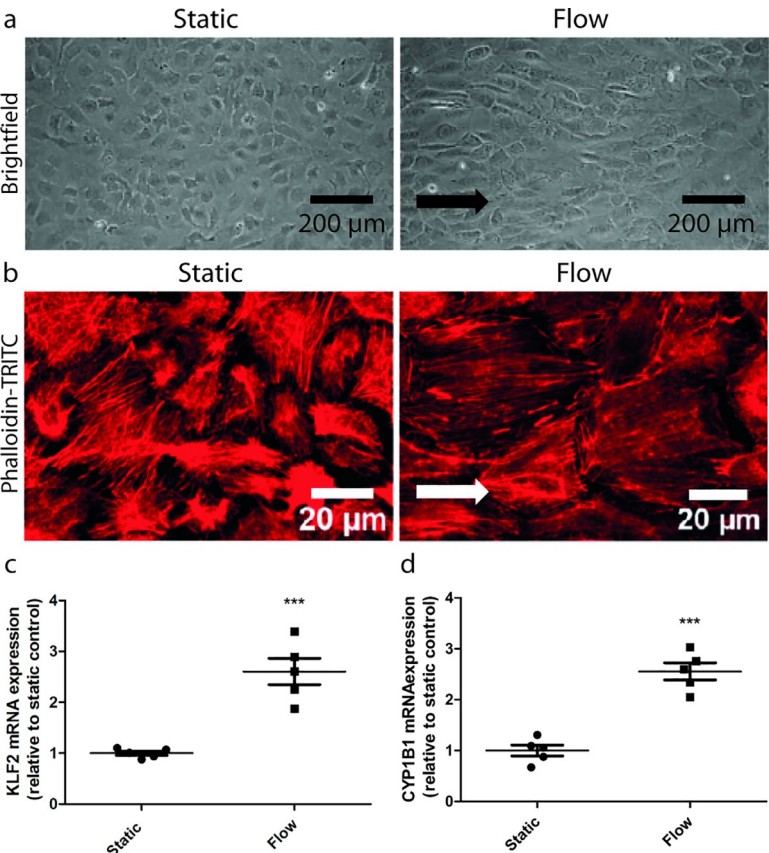

**Fig 1. Laminar flow induces morphological changes and increases expression of key regulators of the shear stress response in GEnC.** mGEnCs were cultured under static conditions or under a laminar flow of 5 dyn/cm$^2$ for 7 days. Representative brightfield images (**A**) and representative fluorescent images of β-actin filaments stained with Phalloidin-TRITC (**B**). Arrows indicate the direction of the flow. mRNA expression levels of the shear stress responsive genes KLF2 (**C**) and CYP1B1 (**D**) in mGEnCs cultured under either static conditions or flow. *** p<0.001.

skeleton (**Fig 1B,** left panel), whereas when cultured under flow, GEnC partially demonstrated actin fibre organization in the direction of the flow (**Fig 1B,** right panel).

## Laminar flow increases expression of key regulators of shear stress response in GEnC

Krüppel-Like Factor 2 (KLF2) and Cytochrome P450 B1 (CYP1B1) are key regulators of the shear stress response in macrovascular endothelial cells and are responsible for activating numerous downstream signalling pathways in response to shear stress [9–11,27,28]. We therefore examined the effect of flow on the expression of KLF2 and CYP1B1 in GEnC, and found that the expression of both KLF2 and CYP1B1 was enhanced in GEnC cultured under flow compared to GEnC cultured under static conditions (p<0.001, **Fig 1C** and **1D**).

## Laminar flow increases NO production by GEnC

NO is a pivotal paracrine factor for maintaining a healthy endothelium and the synthesis of NO is known to be increased in macrovascular endothelial cells in response to laminar flow [29] and in GEnC upon exposure to shear stress for 1 day [16]. We therefore investigated the effect of culturing GEnC under laminar flow for 7 days on NO production by GEnC. NO

production was increased (p = 0.0094) in GEnC upon exposure to shear stress for 7 days (**Fig 2A** and **2B**). The mRNA expression of endothelial Nitric Oxide Synthase (eNOS) was increased when GEnC were cultured under laminar flow (**Fig 2D,** p = 0.0179). Furthermore, eNOS protein expression was also increased when GEnC were exposed to shear stress (**Fig 2G** and **2H,** p = 0.04). Both RNA and protein expression of iNOS were not increased in response to shear stress (**Fig 2C, 2E** and **2F**). Thus, laminar flow seems to increase synthesis of NO by GEnC via increased expression of eNOS.

## Laminar flow induces changes in the glycocalyx of GEnC

The endothelial glycocalyx is essential for glomerular structural integrity. Shear stress has been shown to affect the glycocalyx composition of macrovascular endothelial cells and the expression of glycocalyx modifying enzymes [13–15] Therefore, we addressed the influence of laminar flow on the glycocalyx of GEnC. The lectin WGA is able to detect both HA and HS, whereas JM403 is a monoclonal antibody specific for HS. The expression of HS under flow conditions, as measured with JM403, increased compared to static conditions (**Fig 3A** and **3B,** p = 0.01). The enhanced HS expression was accompanied with a decreased mRNA expression of the HS-degrading enzyme heparanase (HPSE1) (**Fig 3E,** p<0.001). Moreover, the expression of the HS-synthesizing enzyme exostosin-1 (EXT1) was enhanced (**Fig 3F,** p = 0.003), while the expression of two other HS-synthesizing enzymes; exostosin-2 (EXT2) and bifunctional heparan sulfate N-deacetylase/N-sulfotransferase 2 (NDST2) did not change in response to shear stress (**Fig 3G** and **3I**). In addition, the expression of NDST1 was decreased when GEnC were exposed to laminar flow (**Fig 3H,** p = 0.04). (**Fig 3C** and **3D**). The mRNA expression of the HA-biosynthetic genes [30] Hyaluronan Synthase 1 (HAS1) decreased, while HAS2 mRNA increased under laminar flow (**Fig 3J** and **3K,** p = 0.008 and p = 0.02, respectively). mRNA expression of HAS3 was under the detection limit and therefore not shown. Furthermore, the expression of the HA-degrading enzyme hyaluroniase-1 (HYAL1) was enhanced (**Fig 3L,** p<0.001), whereas HYAL2 expression was unaffected by shear stress (**Fig 3M**). Although the expression of HA-synthesizing and HA-degrading enzymes altered, the intensity of the WGA staining (which stains for both HS and HS) did not alter between GEnC cultured under either static conditions or under laminar flow. In summary, laminar flow affects the glycocalyx composition of GEnC, by decreasing the expression of HPSE and the subsequent increased expression of HS.

## Laminar flow induces a quiescent inflammatory and anti-oxidant phenotype in GEnC

Laminar flow induces a quiescent phenotype, characterized by increased expression of anti-oxidant enzymes for enhanced protection against oxidative stress and decreased expression of inflammatory markers in macrovascular endothelial cells [31]. This quiescent phenotype is important for endothelial cells to maintain its physiological functions [11,31]. To examine the effect of shear stress on the anti-oxidant response in GEnC, the expression of Nuclear Factor erythroid 2-related factor (Nrf2) was examined. Nrf2 is a master regulator of the anti-oxidant response and able to increase the expression of the anti-oxidant enzymes heme oxygenase 1 (HO-1) and NAD(P)H dehydrogenase quinone 1 (NQO1). Laminar flow increased the expression of Nrf2 in GEnC (**Fig 4A,** p = 0.02). In addition, a strong trend in increased expression of HO-1 could be observed (**Fig 4B,** p = 0.05) and the expression of NQO1 was elevated in response to shear stress (**Fig 4C,** p = 0.004).

Next, the influence of shear stress on the inflammatory phenotype of GEnC was determined. Laminar flow resulted in decreased expression of the key endothelial inflammatory

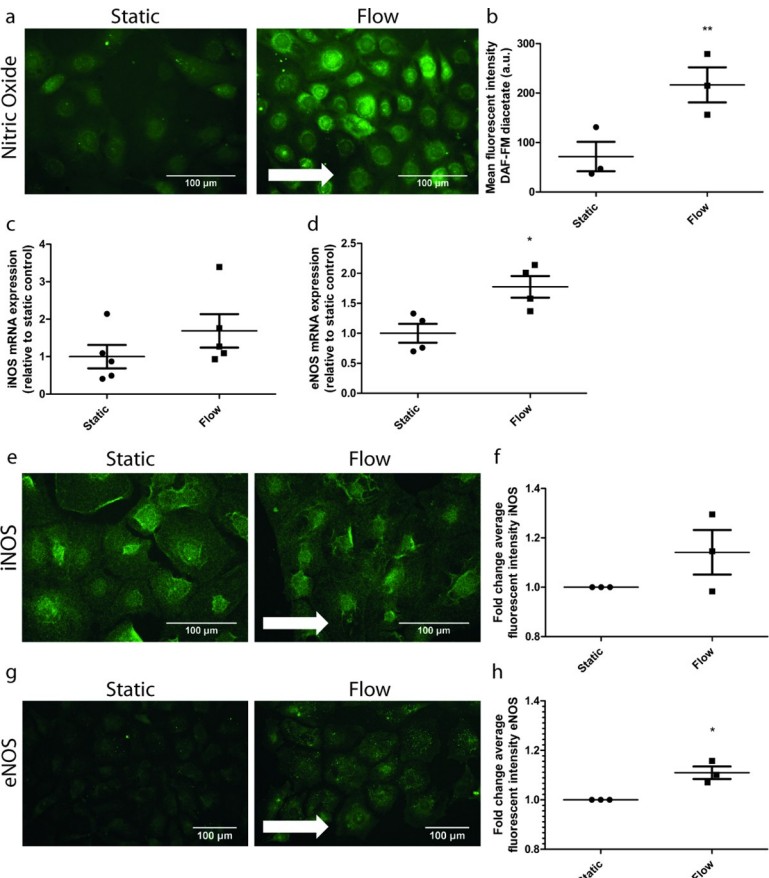

**Fig 2. Laminar flow increases NO production in GEnCs.** mGEnCs were cultured either under static conditions or under a laminar flow of 5 dyn/cm2 for 7 days. Representative images of stainings for NO (probed with DAF-FM diacetate) (**A**), iNOS (**E**) and eNOS (**G**) of mGEnC cultured under static (left panel) or flow (right panel). Arrows indicate the direction of the flow applied. Quantification of the fluorescent intensity of NO (**B**), iNOS (**F**) and eNOS (**H**) staining, respectively. mRNA expression levels of iNOS (**C**) and eNOS (**D**) in mGEnCs cultured under either static conditions or flow. * $p < 0.05$ ** $p < 0.01$.

marker ICAM-1 (**Fig 4D,** p<0.001). We subsequently investigated if GEnC cultured under flow also responded differently to inflammatory conditions compared to GEnC not exposed to shear stress. ICAM-1 expression increased ~3-fold in GEnC cultured under static conditions, while iNOS expression remained unaltered, upon exposure to 10ng/mL TNFα (**Fig 4E** and **4F**, p<0.05). When GEnC were exposed to shear stress, ICAM-1 expression increased ~25 fold upon TNFα exposure (**Fig 4E**, p<0.001). Furthermore, in contrast to GEnC cultured under static conditions, TNFα exposure resulted in an increased iNOS expression in GEnC cultured under laminar flow (**Fig 4F,** p<0.001). In summary, laminar flow leads to enhanced expression of anti-oxidant enzymes and an altered inflammatory state of GEnC.

## Discussion

The present study is the first to detail the effect of laminar flow on GEnC in culture. We demonstrated altered morphology of GEnC, increased expression of shear stress-responsive genes, elevated NO production, and an altered anti-oxidant and inflammatory response. Furthermore, shear stress altered the glomerular endothelial glycocalyx, characterized by increased HS expression.

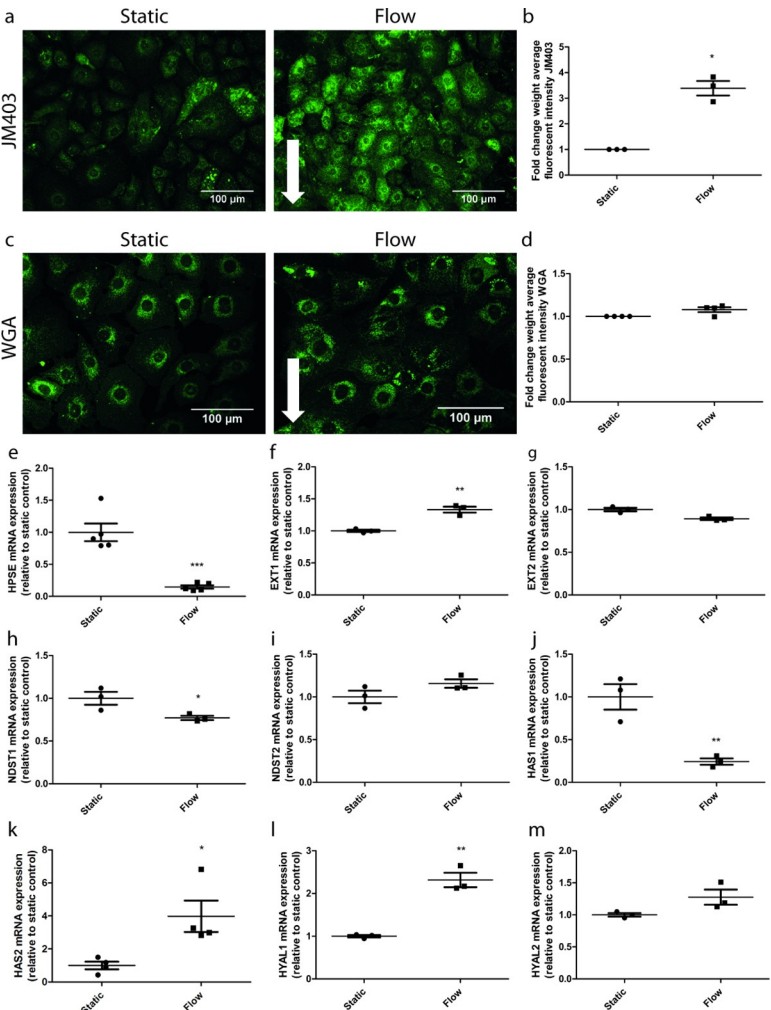

**Fig 3. Laminar flow alters the composition of the endothelial glycocalyx.** mGEnCs were cultured under static conditions or under a laminar flow of 5 dyn/cm2 for 7 days. Representative images for the HS staining with JM403 (**A**) or WGA (**C**) under static (left panel) or flow (right panel). Arrows indicate the direction of the flow applied. Quantification of the fluorescent intensity of the JM403 (**B**) and WGA (**D**) staining. mRNA expression levels of HPSE (**E**), EXT1 (**F**), EXT2 (**G**), NDST1 (**H**), NDST2 (**I**), HAS1 (**J**), HAS2 (**K**), HYAL1 (**L**) and HYAL2 (**M**) in mGEnCs cultured under either static conditions or flow. * $p < 0.05$ ** $p < 0.01$ *** $p < 0.001$.

Chronic exposure to physiological shear stress induces a quiescent inflammatory phenotype in macrovascular endothelial cells [8–11] and prevents TNFα-induced pro-inflammatory responses in the aorta [32]. We found in the current study that GEnC exposed to shear stress also showed a quiescent inflammatory phenotype, as evident from decreased ICAM-1 expression. However, GEnC were more responsive under inflammatory conditions when exposed to shear stress, compared to GEnC cultured under statistic conditions. Additional findings in our study highlight how GEnC respond differently to shear stress compared to macrovascular endothelial cells. A recent study showed the increased expression of NDST1 and EXT2 upon exposure to shear stress for 7 days in macrovascular endothelial cells [33]. Furthermore, a more intense WGA staining was observed in the macrovascular endothelium as a result of laminar flow for 7 days [33]. In the present study, however, we detected a decreased NDST1 expression and were unable to show an effect of shear stress on EXT2 expression and WGA

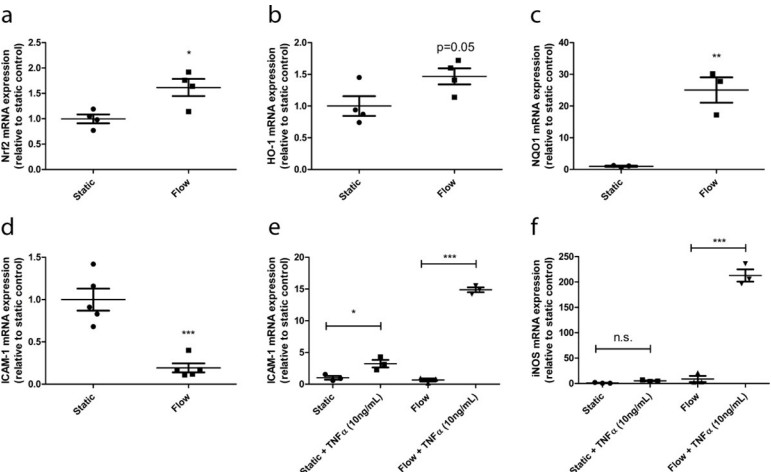

**Fig 4. Laminar flow increases the expression of anti-oxidant genes and decreases the inflammatory basal state of GEnC.** mGEnC were cultured under static conditions or under a laminar flow of 5 dyn/cm2 for 7 days. mRNA expression levels of Nrf2 (**A**) HO-1 (**B**), NQO1 (**C**) and ICAM-1 (**D**) in mGEnCs cultured under either static conditions or flow. mRNA expression levels of ICAM-1 (**E**) and iNOS (**F**) in mGEnCs cultured under static conditions and under flow, either unstimulated or stimulated with 10ng/mL TNFα for 18hrs * p<0.05, ** p<0.01, *** p<0.001.

staining. Taken together this suggests that GEnC respond differently to laminar than macro-vascular endothelial cells. We therefore do not recommend to extrapolate findings about the influence of laminar from studies using macrovascular endothelial cells to the glomerular endothelium. More studies are therefore required to fully characterize the influence of laminar flow on GEnC, for example by studying the effect of a disturbed glomerular blood flow on the inflammatory phenotype of GEnC. GEnCs cultured under static conditions are often used as an experimental tool in research on the glomerular filtration barrier [34–36]. Based on the findings in the current study, culturing GEnC under laminar flow for 7 days might improve the potential of GEnC to mimic glomerular physiology *in vitro* and eventually a better under-standing of the glomerular filtration barrier.

The endothelial glycocalyx is crucial for regulating kidney function, for example by main-taining glomerular structural integrity and contributing to the endothelial barrier function [3–5]. In this study, we discovered that shear stress altered the glycocalyx composition of GEnC. This was evident by the increased expression of HS, the main functional component of the endothelial glycocalyx [37]. The increased expression of HS correlated with the decreased expression of HPSE when GEnC were cultured under laminar flow. Importantly, HPSE is also crucial for the development of acute experimental glomerulonephritis and diabetic nephropa-thy [7,38]. The induction of experimental glomerulonephritis resulted in increased HPSE expression, a decreased glomerular HS expression and increased proteinuria in wildtype mice. In HPSE-deficient mice, however, renal function was improved and proteinuria was reduced compared to wildtype mice upon inducing experimental glomerulonephritis [38]. Further-more, a systemic HPSE knock-out prevented mice from the development of proteinuria upon inducing experimental diabetic nephropathy [7]. HPSE expression is also influenced by shear stress in the macrovasculature; HPSE expression and activity were increased in aortic regions with a disturbed aortic blood flow [13]. These aortic regions showed more advanced progres-sion of atherosclerotic lesions and increased inflammation compared to regions with a normal blood flow. Combined with our finding in the current study about shear stress-mediated downregulation of HPSE in GEnC, this might suggest that a normal glomerular blood flow is

an important factor to prevent HPSE upregulation, the preservation of a quiescent endothelial phenotype and the prevention of glomerular inflammation.

A limitation of the current study is that monocultures of conditionally immortalized GEnC were used. Extensive efforts are currently underway in the development of glomerulus-on-a-chip models in order to recapitulate the *in vivo* situation by co-culturing GEnC with podocytes [39–41]. In these devices, GEnC should be cultured under laminar flow, as our study shows that culturing GEnC under physiological shear stress has a profound effect on their morphology and functioning. Studies investigating the specific effect of flow on GEnC in these devices, however, are still lacking. It is therefore impossible to dissect the influence of flow and the impact of co-culture with podocytes on the phenotype of GEnC in these devices. More detailed information about the individual contributions of the co-culture and laminar flow to the glomerulus-on-a-chip models might eventually lead to a better understanding of the physiology of the *in vivo* glomerulus.

In conclusion, culturing GEnC under laminar flow for 7 days has a profound effect on their morphology and functional phenotype. In addition, GEnC respond differently to shear stress than macrovascular endothelial cells, as evidenced by the response of GEnC under inflammatory conditions and alterations in glycocalyx composition. Culturing GEnC under laminar flow for 7 days might therefore be a new necessary *in vitro* tool for a better understanding of glomerular (patho)physiology *in vivo*.

## Acknowledgments

The authors want to thank Marieke Willemse for excellent technical support.

## Author Contributions

**Conceptualization:** Daan C. 't Hart, Johan van der Vlag, Tom Nijenhuis.

**Formal analysis:** Daan C. 't Hart.

**Funding acquisition:** Johan van der Vlag, Tom Nijenhuis.

**Investigation:** Daan C. 't Hart.

**Project administration:** Johan van der Vlag, Tom Nijenhuis.

**Supervision:** Johan van der Vlag, Tom Nijenhuis.

**Visualization:** Daan C. 't Hart.

**Writing – original draft:** Daan C. 't Hart.

**Writing – review & editing:** Johan van der Vlag, Tom Nijenhuis.

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
