## [Decision Letter · Decision Letter 0]

21 Apr 2021

Laminar flow substantially affects the morphology and functional phenotype of glomerular endothelial cells

PONE-D-21-07368

Dear Dr. van der Vlag,

We’re pleased to inform you that your manuscript has been judged scientifically suitable for publication and will be formally accepted for publication once it meets all outstanding technical requirements.

Kind regards,

Nicole Endlich, Prof

Academic Editor

PLOS ONE

1. In your Methods section, please provide additional details regarding the cell lines used in your study and ensure you have described the source. For more information regarding PLOS' policy on materials sharing and reporting, see https://journals.plos.org/plosone/s/materials-and-software-sharing#loc-sharing-materials, and for more information on PLOS ONE's guidelines for research using cell lines, see https://journals.plos.org/plosone/s/submission-guidelines#loc-cell-lines.

Reviewers' comments:

Reviewer's Responses to Questions

**Comments to the Author**

1. Is the manuscript technically sound, and do the data support the conclusions?

Reviewer #1: Yes

Reviewer #2: Yes

2. Has the statistical analysis been performed appropriately and rigorously? 

Reviewer #1: Yes

Reviewer #2: Yes

3. Have the authors made all data underlying the findings in their manuscript fully available?

Reviewer #1: Yes

Reviewer #2: Yes

4. Is the manuscript presented in an intelligible fashion and written in standard English?

Reviewer #1: Yes

Reviewer #2: Yes

5. Review Comments to the Author

Reviewer #1: In their very well designed comprehensive study the Authors present novel findings on the effects of a long-term shear stress on glomerular endothelial cells in culture. The Authors have investigated and characterized the key GEnC features that are typically sensitive to the shear stress, such as NO synthesis, expression of stress-responsive genes and proteins, or cytoskeleton shape.

Presented results are convincing and they clearly indicate that cells in culture should be exposed to stimuli similar to those in vivo. Otherwise, the results may not sufficiently reflect the actual cell reaction.

On the other hand, this is a very inspiring study that should be (and I believe, will be) followed by next experiments investigating the reciprocal contacts between the GEn cells and the podocytes , in conditions where both cell types are grown under laminar flow.

Reviewer #2: In my view, it is a flawless study. No suggestions for improvements

This is a well-done study on an important subject. Such a study has been overdue. I recommend to accept this manuscript without any changes. In my view, it is a flawless study.

6. PLOS authors have the option to publish the peer review history of their article (what does this mean?). If published, this will include your full peer review and any attached files.

Reviewer #1: **Yes: **Barbara Lewko, PhD, Department of Pharmacological Pathophysiology, Faculty of Pharmacy, Medical University of Gdansk, Poland

Reviewer #2: No

---

## [Editor Report · Acceptance letter]

27 Apr 2021

PONE-D-21-07368 

Laminar flow substantially affects the morphology and functional phenotype of glomerular endothelial cells 

Dear Dr. Nijenhuis:

I'm pleased to inform you that your manuscript has been deemed suitable for publication in PLOS ONE. Congratulations! Your manuscript is now with our production department. 

Kind regards, 

on behalf of

Professor Nicole Endlich 

Academic Editor

PLOS ONE